# Towards concurrent real-time audio-aware agents with deep reinforcement learning

Anton Debner[*1] and Vesa Hirvisalo[1]

[1]Aalto University
{anton.debner, vesa.hirvisalo}@aalto.fi

## Abstract

Audio holds significant amount of information about our surroundings. It can be used to navigate, assess threats, communicate, as a source of curiosity, and to separate the sources of different sounds. Still, these rich properties of audio are not fully utilized by current video game agents.

We use spatial audio libraries in combination with deep reinforcement learning to allow agents to observe their surroundings and to navigate in their environment using audio cues. In general, game engines support rendering audio for one agent only. Using a hide-and-seek scenario in our experimentation we show how support for multiple concurrent listeners can be used to parallelize the runtime operation and to enable using multiple agents. Further, we analyze the effects of audio environment complexity to demonstrate the scalability of our approach.

## 1 Introduction

Modern video games have rich and high-quality audio scenes. Such audio holds significant amount of information on the virtual environment. Deep learning is becoming increasingly popular as an approach to extract information from virtual environments but is not fully utilized to its potential, especially when audio is considered.

For video game non-player characters (NPCs), the utilization of audio could support intuitive, human-like behavior. For example, an audio-aware agent could have a fear of loud noises as a self-preservation mechanic. Audio-aware agents could be curious about odd noises, gather information about the surrounding environment, and become susceptible to audio-based distractions deliberately made by human players or even other NPCs.

While using deep learning for audio sensing in video games is not yet an extensively researched topic, there exists a wide range of applications that utilize it. For example, audio has been used to classify gunshot noises in video games [1], to enhance exploration during training [2], and as a generic addition to visual observations [3]. Games engines can integrate multi-disciplinary functionalities and have wide applicability beyond entertainment. As such, game engines can be seen as generic platforms [4] for research on deep learning methods.

However, game engines are often created with the assumption that audio is rendered for the human player only. As such, environments can have only one audio listener at a time. This listener can be assigned either to the player or a single audio-aware agent. In order to support multiple concurrent audio-aware agents, advanced methods are needed.

In this paper, we describe our experimentation utilizing deep reinforcement learning (DRL). Our experimentation focuses on sound source localization and is based on a hide-and-seek scenario, where agents must locate targets based on spatial audio cues. Our main contribution is to show that game engines can be enhanced by using multi-listener audio rendering to enable parallelization of the runtime operation and use multiple agents. We demonstrate how spatial audio libraries can be used in a popular game engine with machine learning support (the Unity engine) to construct a multi-listener based system that can be run in real-time on modern workstations and yield good performance even if the audio environments are complex. Our methodology is not limited to games. It can be used in other domains that use similar tools for creating virtual environments and have similar requirements. Our code is available on GitHub[1].

The structure of this paper is the following. We begin our presentation by reviewing the related work in Section 2. In Section 3, we describe our methodology and our selection of tools. We continue by describing our experimental setup (Section 4) and the results of our experiment (Section 5). We end our paper with a short discussion (Section 6) and our conclusions (Section 7).

## 2 Related work

Deep reinforcement learning (DRL) [5] is a branch of machine learning, where agents are trained in a trial-and-error manner. The training is done in an environment, where learning is based on the rewards the agents receive as feedback on their actions. As the agents are truly run during the training, making

---

[*]Corresponding Author.

[1]https://github.com/Aalto-ESG/aaaa-2025

Proceedings of the 6th Northern Lights Deep Learning Conference (NLDL), PMLR 265, 2025.

observations in the training environment and overall performance of the training steps are essential.

There exists a vast amount of machine learning literature on audio. Considering our application area, Grumiaux et al. [6] survey different deep learning methods in general for sound source localization. The survey from Latif et al. [7] focuses only on the use of DRL but considers a wide range of audio-based applications. Beig et al. [8] review spatial sound rendering for virtual environments and games.

In applying DRL for game applications, timing behavior is central in addition to the related computational efficiency. Hedge et al. [3] study the performance of different model architectures in a series of tasks that require the agent to recognize sounds. They decouple audio rendering from dedicated sound hardware to enable faster-than-realtime parallel simulation in Vizdoom [9], resulting in faster training of audio-based agents. Their work is an example of agents learning to play a game, instead of learning to act as NPCs. They reach a training throughput of 120 000 samples per second with audio and 150 000 samples per second without audio.

Cowan et al. [10] present a computationally efficient audio rendering system for game NPCs. They also recognize the problems caused by poor utilization of spatial audio by NPCs. Their main idea is that if the audio for the player is rendered using a graph-based sound propagation method, the generated graph can immediately be reused for NPCs with low additional computational cost.

Environments including the interaction by the agents is typically modeled or simulated in a manner that captures the related physical realism. Chen et al. [11] present SoundSpaces 2.0, a simulator for training audio-based agents in environments modeled after the real world. Gan et al. [12] show their results with discrete in-door navigation using audiovisual data with AI2-THOR platform. They create an offline dataset from AI2-THOR platform running on Unity game engine.

# 3 Audio agents in virtual environments

In this section, we describe our methodology and the related selection of tools. We begin by discussing the problem space, and continue by describing our platform selection, the related observation tools, and the audio rendering libraries we have selected. After these, we describe our approach to multiple listeners, which are the key aspects of this research.

## 3.1 The problem space

While real-world embodied agents and video game agents can utilize audio using similar methods, they have a different set of requirements.

The audio-aware agents should be able to utilize audio in a useful way, but their output does not need to be perfect. Unlike with real-world embodied agents, mistakes can even be beneficial for the gameplay experience. Additionally, the agents need to be able to work with soft real-time simulations. Missing a deadline is not critical, but can harm the experienced quality of service. The agents also do not need to make decisions on every simulation tick, but still often enough to be able to fluently navigate the simulated environment.

Considering the use of deep reinforcement learning for the audio-aware agents there are two challenges: 1) The computational requirements of audio-aware agents and the related spatial audio rendering by the runtime system should be relatively low. Running in real-time also during training avoids audio distortions. 2) The training of the agents should be efficient enough in order to make their use realistic in the context of video game development. Especially the latter aspect is dependent on the operating environment of the agents as complex audio scenes make the training challenging.

## 3.2 Platform

We use Unity [4] as our platform. We combine Unity with its deep reinforcement learning framework ML-Agents [4]. ML-Agents includes a training pipeline for simple DRL architectures with popular methods, such as PPO [13]. It also exposes the Unity API for external tools, such as Ray RLlib [14], enabling the use of complex DRL models.

## 3.3 Observations

Virtual agents observe their surroundings through abstractions called sensors. In the context on ML-Agents, the simplest sensors are fixed length vector sensors, that can be filled with arbitrary values.

To be able to use audio data as the input for our policy network, we need to have an audio sensor, but ML-Agents does not include any audio-related sensors. We opted to use a third-party sensor from GitHub user mbaske [15], which is available on GitHub with the MIT license. The sensor supports observing stereo audio in spectral domain using Short-Time Fourier Transform (STFT) [16] with different windowing functions, such as Hanning and Rectangular window. The windowing function affects the accuracy of the amplitude and frequency of the sensor. The sensor constructs an observation batch by sampling the audio into a buffer over several simulation ticks.

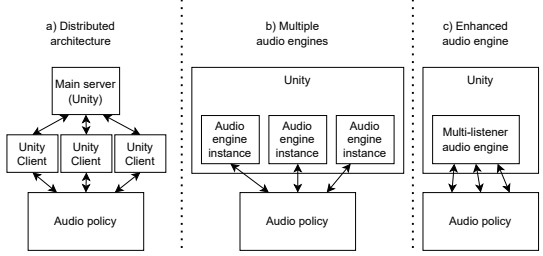

**Figure 1.** Approaches for multi-listener setup.

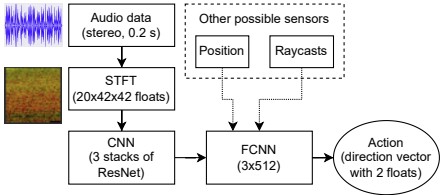

**Figure 2.** Sensor data processing in our agent design.

**Table 1.** System specifications

| Label | CPU | GPU | RAM | OS |
|---|---|---|---|---|
| A | Intel 13900k | GTX 1080 ti | 64 GB | Windows 11 |
| B | Intel 13400 | RTX 3080 | 32 GB | Ubuntu 22.04 |

## 3.4 Audio rendering libraries

By default, popular game engines such as Unity and Unreal Engine support simplified spatialization without accounting for reflections or occlusions. To enable more complex audio observations, we use Steam Audio [17] as a plugin for Unity.

Steam Audio is one of the several libraries [8] that supports spatial audio. It supports spatialization through head-related transfer function (HRTF), which models the interaural level differences (ILD) and the interaural time difference (ITD). It also supports occlusions and reflections either by real-time ray tracing for dynamic scenes or baked audio propagation for static scenes.

## 3.5 Multi-listener support

Even with Steam Audio, Unity supports only one active audio listener at a time. This means that all audio will be rendered from the point of view of a single agent. From the training perspective, having only one audio listener limits parallelization of the training process as one instance will only support one agent. From the runtime perspective, the audio listener is often reserved for the player.

We propose three basic ways to enable multi-listener simulation. These are shown in Figure 1.

1. Each client has its own audio engine instance running and will therefore be able to feed multiple agents with audio observations. The game logic runs on the main server, but the audio events are replicated on the remote clients.

2. Creating multiple instances of the audio engine within a single game instance, thus removing the overhead caused by the network.

3. An audio engine that natively supports multiple listeners, which eases internal optimizations.

Integrating the multi-listener support directly into the engines opens optimization opportunities and removes overhead but also increases significantly the implementation effort. We selected Option 1 for our experimentation.

# 4 Experiment setup

In this section, we outline our experiment setup. Our goal is to evaluate the effect of using multiple listeners as proposed in Figure 1 (leftmost), while also training and evaluating a DRL-based audio-aware agent using Unity and Steam Audio. Multi-listener measurements are performed on hardware configuration A as seen in Table 1. The agents are trained using both configurations A and B.

We consider a hide-and-seek scenario, where an audio-aware agent attempts to find the target (e.g., the player) in an indoor area. The target is constantly making noise by randomly playing one of eight footstep audio clips. In order to simplify the evaluation of the agents, the target is represented as a static object that does not move around in the environment. The motivation for this setup comes from a typical scenario, where a player attempts to evade detection.

The agent can observe its surroundings with sensors as described in Figure 2. The audio data are fed through STFT to a convolutional neural network (CNN), and the integration of other sensor data is done with a fully connected neural network (FCNN). In our measurements, the other sensors are excluded, and the audio sensor is the only source of data.

## 4.1 Multi-listener setup

We measure the impact of running multiple simulation instances in parallel as proposed in Figure 1 (leftmost). Each Unity instance runs the exact same scene with Steam Audio and one audio agent with a CNN-based audio model. The number of audio sources is varied between 1, 10 and 30 audio sources to evaluate the impact of having several audio sources.

## 4.2 Environments

Agent performance is compared in multiple environments with increasing complexity. The easiest

environment ("Simple") is an empty hall without obstructions. The second environment ("Medium") adds some obstructions in the form of rooms and walls, making it necessary to utilize indirect audio reflections. The third environment ("Complex") increases the difficulty by including more walls. The environments are described in more detail in Appendix B.

## 4.3 Agent design

The agents are trained using PPO [13] over 10 million samples in the second ("Medium") environment. The training hyperparameters are listed in the Appendix A.1.

The policy design is shown in Figure 2. In order to isolate the usefulness of the audio observations, the audio sensor is the only sensor included in the experiments. The agent outputs two values, representing the x and y values of a 2-dimensional vector.

The reward signal, denoted as $R$, is calculated by taking a dot product between the direction of the shortest path to the target and the action vector. To encourage finishing episodes as quickly as possible, we convert the result into a penalty by shifting it from range [-1, 1] to range [-2, 0]. Finally, it is scaled by the fixed delta time to make it invariant to the simulation physics update frequency. The agent is given an additional reward of 2 upon finishing an episode.

The reward $R$ at physics update $n$ can be expressed as:

$$R_n = ((\mathbf{a} \cdot \hat{\mathbf{s}}) - 1)\Delta t + \begin{cases} 2, & \text{if target reached} \\ 0, & \text{otherwise} \end{cases} \quad (1)$$

where $\mathbf{a}$ is the action vector, $(\hat{\mathbf{s}})$ is a unit vector pointing to the shortest path to target and $\Delta t$ is the time between two physics updates (called fixed delta time in Unity). The maximum achievable reward during an episode is 2.

To prevent agents from getting stuck between two locations, they need information on locations already visited. Instead of using more complex neural networks such as RNNs [18] or LSTMs, we used a simple weighted navigation grid (see Appendix A.3). The grid node weights indicate the likelihood of audio coming from that direction. We adjust these weights based on neural network outputs and navigate towards the highest-weighted neighboring node. If there are no occlusions between the target and the agent, the navigation grid is ignored, and the agent will travel directly towards the output direction.

## 4.4 Evaluation metrics

We evaluate the viability of our multi-listener approach by measuring the CPU and memory (MEM)

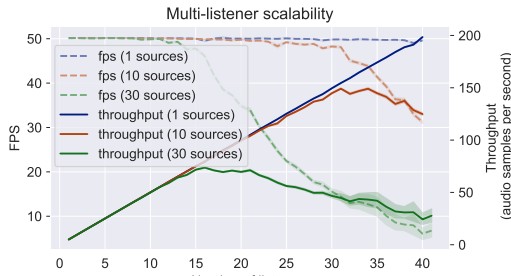

**Figure 3.** Main server FPS and total audio sample throughput. Each audio sample contains a 0.2 second audio buffer collected over 10 simulation ticks.

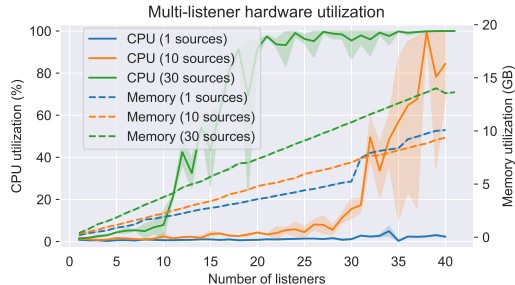

**Figure 4.** Average hardware utilization. CPU utilization is an average over all 24 cores. Both metrics are also affected by background processes.

utilization with different configurations.

We measure agent performance using sample throughput, training time, and cumulative training reward during training. After training, we evaluate the agent with a fixed set of 100 randomized episodes in each environment using the SPL metric [19].

SPL is defined as

$$\text{SPL} = \frac{1}{N} \sum_{i=1}^{N} S_i \frac{l_i}{\max(p_i, l_i)} \quad (2)$$

where N is the number of episodes, $S_i$ is 1 if the episode $i$ succeeded and 0 otherwise. The $l_i$ is the shortest path to target and $p_i$ is the length of the path taken by the agent.

As such, SPL accounts for both the failure to reach the goal and the optimality of the path compared to the shortest path. It is better than simply measuring the number of steps or time to reach the target, since the distance to the target varies by episode. However, since SPL relies on knowing the optimal path length, it is not suitable for moving targets as such.

## 5 Results

In this section, we describe our measurement results on multi-listener performance, training performance,

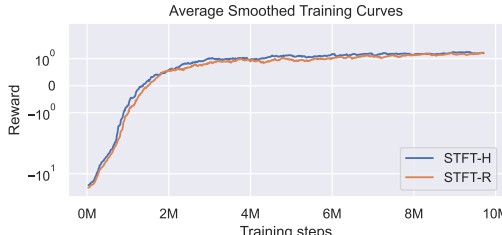

**Figure 5.** Training reward over time.

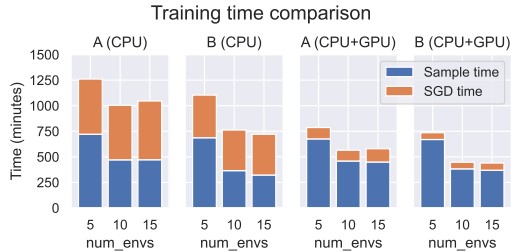

**Figure 6.** Training time comparison.

## 5.1 Multi-listener performance

Figure 3 shows how performance scales with parallel audio-aware agents in three different configurations. The figure shows the simulation frames per second (FPS) processed by the main server, which is set at a fixed rate of 50 FPS. Values lower than 50 mean that the simulator can no longer keep up with real time. With only one audio source, the FPS remains nearly stable for up to 40 listeners. With 30 sources, the simulation supports roughly 12 concurrent audio-aware agents.

Figure 3 also shows the total sample throughput for all audio-aware agents. Each audio sample is computed from a 0.2 second long audio buffer, which requires 10 simulation frames at 50 FPS. Sample throughput often correlates directly with the time required to train a DRL-based agent. From the figure, we can see that the sample throughput increases linearly to 200 audio samples per second (SPS) with one audio source. With 30 audio sources, the throughput caps at 75 SPS with around 15 environments.

Figure 4 shows how the utilization of the CPU scales with parallel Unity instances. With one audio source, CPU utilization stays below 5 % even at 40 parallel instances. With 10 audio sources, CPU utilization stays reasonably low until 25 environments before increasing exponentially. With 30 audio sources, CPU usage increases already after 10 instances. At 41 parallel instances, the entire system becomes unresponsive.

Figure 4 also shows that each instance has a near linear effect on memory usage. However, the number of audio sources affects the coefficient of this linear behavior. With 30 audio sources, the setup requires nearly double the memory compared to having one source. With 30 audio sources, one instance consumes roughly 300 MB of memory. The sudden increase in memory usage with one audio source after 30 listeners is likely caused by background load.

## 5.2 Training performance

We trained two DRL-based agents using two different STFT window functions (STFT-R for rectangular and STFT-H for Hanning window). The training was repeated three times. During training, we collect a sample on every simulation step. This increases the sampling throughput, but the samples will have overlap with each other as the audio buffer holds data over 10 steps.

Figure 5 shows the training reward curves. The curves are averaged over all three runs and then smoothed. The reward curves converge quickly and there is little improvement after 5 million steps.

Figure 6 shows the total training time for 10 million steps in four configurations. In addition to the total time, it shows the time spent collecting samples from agents and the time spent updating the model with the Stochastic Gradient Descent (SGD) [13]. Increasing the number of parallel Unity instances to 10 significantly increases sample throughput. Additionally, using a GPU nearly halves the training time by reducing the time spent on SGDs.

Despite having a higher-end CPU, configuration A (CPU) is slower than configuration B (CPU). This could be due to the different operating system or some other differences in hardware details, such as the memory speed or motherboard configuration.

In conclusion, sample throughput is a clear bottleneck in a GPU-accelerated system. It might be possible to further increase the sample throughput by having multiple agents in one Unity instance and by having faster-than-realtime audio rendering. Both of these could reduce the overhead from running multiple Unity instances in parallel.

## 5.3 Audio-aware agent performance

After training the STFT-R and STFT-H agents, we compared them to a random agent that randomly traverses the environment. The results are averaged over three training-evaluation cycles.

Figure 7 compares the SPL values achieved by different agents, where the best possible value is 1. As expected, the audio-aware agents perform best in a simple scene without obstacles, reaching an SPL of around 0.85. The STFT-H seems to perform

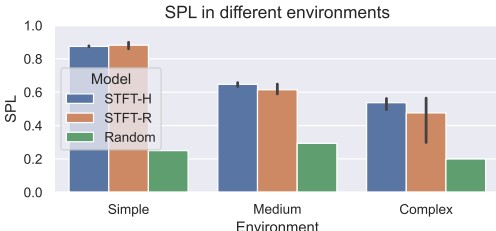

**Figure 7.** Measured SPL in different testing environments. Higher values are better.

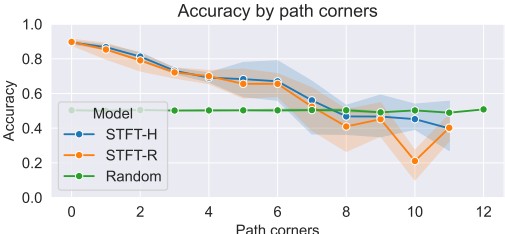

**Figure 8.** Accuracy by remaining shortest path corners.

slightly better than SRFT-R overall. When obstacles are added, the agents perform worse, but still significantly better than the random agent. As reference, Anderson et al. [19] mention that an SPL of 0.5 shows a good level of navigation performance in reasonably complex previously unseen environments.

Figures 8 and 9 show how the remaining distance from the agent to the target affects agent prediction accuracy. If we denote the accuracy as $A \in [0, 1]$, then $A = 1$ means that the action-vector is aligned with the shortest path to target. Likewise, $A = 0$ means that the action-vector deviates from the direction of the shortest path by 180 degrees.

Figure 8 shows the accuracy based on how many corners the shortest path to target has. The number of corners depends on the obstacles between the agent and the target, since the agent must turn at least twice to get around an obstacle. Figure 9 shows the shortest path length in Unity units. From these figures, we can see that the remaining distance has a more linear effect on accuracy, while the amount of remaining obstacles has a more drastic effect.

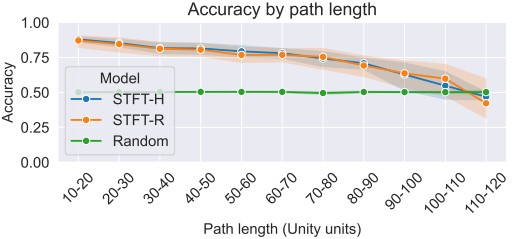

**Figure 9.** Accuracy by remaining shortest path length.

## 6 Discussion

The lack of support for multiple audio listeners is a significant drawback in most current game engines, as it limits the options of using audio as an observation for deep learning agents.

However, our approach of using multiple clients can ease these limitations. Our experimentation demonstrates that with a rather simple workaround in Unity, nearly a dozen audio-aware agents can be supported. However, resource-intensive games probably need more game-specific optimization than what we have done in our experimentation.

Our results also highlight that minor details, such as the STFT windowing function, can noticeably affect agent behavior. Therefore, the audio properties of the environment and the audio sensor must be chosen carefully.

As future work, we suggest looking at the other proposed approaches for creating multi-listener support to reduce overhead. Having only one game instance would remove the need for synchronization between the clients and reduce memory usage. Similar to Cowan et al. [10] it is also likely that some of the audio propagation computation could be reused.

Also, modifying the audio engine to support faster-than-realtime playback would be helpful. For example, in our experiment, the sample throughput was around 250 SPS. This is slow compared to the 120 000 SPS reached with the faster-than-realtime solution by Hegde et al. [3] for VizDoom. Similar to the solution by Hedge et al., the audio backend would likely need to be modified to enable full software rendering of the audio at any sampling rate.

Finally, in addition to performance enhancements, we suggest creating a rich game-oriented benchmark suite for audio-aware agents. Such benchmark suites (e.g., dm_control [20]) enable standardized comparisons of different DRL-based approaches.

## 7 Conclusion

In this paper, we presented our experimentation on using multi-listener rendering in game engines for enabling concurrent audio-aware agents with deep reinforcement learning. Our work demonstrates how current audio libraries can be utilized to create audio-aware agents with reasonable training results.

While our results show the viability of using deep learning based audio-aware agents in video games, our research also highlights the need for further research on the methodology and development of applications and tools. The ability of faster-than-realtime audio operations could significantly speed up training of audio-aware agents. Likewise, native support for multiple audio listeners would make training and runtime more efficient while enabling the use of multiple agents.

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

# A   Agent details

## A.1   Training hyperparameters

The agents are trained using Unity ML-Agents library with PPO. The structure of the neural network (NN) follows the default implementation included in ML-Agents (version 0.30.0) with the hyperparameter setting given in A.1. The audio data are included as a set of images, which ML-Agents feeds to a Resnet-style CNN. The CNN output is concatenated with the vector inputs of other sensors and fed to a linear network. In the experiments presented in this paper, the other sensors were excluded and the agents were using the audio sensor only.

## A.2   Random agent

The random agent does not have any NN-based model. Instead, it chooses random coordinates in the environment and navigates towards them using the Unity NavMesh. Upon reaching these coordinates, it selects new coordinates. This process is repeated until the agent gets within 15 units of the target.

When the agent arrives within 15 units of the target, we assume that it can see the target, and the agent navigates directly towards the target. Without having this kind of detection range, it would be unlikely for the agent to ever encounter the target. The environments are 100x100 units large.

## A.3   Navigation grid

Without any assistance, DRL-based agents can easily get stuck between two points. Typically, this happens near walls: an agent infers that the audio is coming from the direction of the wall. The agent gets stuck in a loop of retreating and approaching. The behavior can be corrected by keeping a record of historical actions and observations, which we implement with a navigation grid (Navgrid).

Navgrid is a grid of evenly spaced navigation nodes placed across the environment. Adjacent nodes are neighbours to each other, if there is a clear unobstructed in line of sight between the nodes. Instead of directly using the NN output for navigation, we use it to update the Navgrid weights and navigate towards the highest-weight neighboring node. When a node is reached, the agent starts navigating towards the neighboring node that leads to the path of highest cumulative discounted weights.

Each node has a weight $W \in [0, 100]$. To represent the uncertainty of old weights, all node weights decay towards a neutral value $W_{neutral} = 20$ at the rate of $\gamma = 0.1$ units per physics update.

The pseudocode for the weight update algorithm is visible in Listing 1. The algorithm updates the weights of the nearby nodes at the maximum depth of $d_{max} = 2$ nodes. The nodes in the general direction of the action will receive an increase in weight, while the nodes in the opposite direction will get a reduction in weight. The change in weight is propagated to further neighbors with a discount of $\lambda^{d-1}$ (where $d$ is the depth and $\lambda = 0.5$) up to the maximum depth $d_{max}$, as the uncertainty of the audio direction increases the deeper we progress into the graph.

The weight of the node closest to the agent is set to 0, as we can be certain that the target is not nearby. This is because the agent does not use the Navgrid, if it has line-of-sight to the target. In our experiment, the agent is given the line-of-sight information, but it could also be predicted by the agent from sensor data.

Figure B.1 shows a Navgrid example with a red line visualizing the current agent output. The small white square near the end of the red lines is the target. Based on previous outputs from the agent, the related heatmap shows the likely location of the target (the higher weights are brighter and red than the lower ones). The black areas in the heatmap are missing values caused by how the heatmap is rendered.

| Parameter | Value |
| --- | --- |
| maximum steps | 10 000 000 |
| batch size | 1024 |
| buffer size | 10240 |
| learning rate | 0.0003 |
| beta | 0.005 |
| epsilon | 0.2 |
| lambd | 0.90 |
| num epoch | 3 |
| hidden units | 512 |
| num layers | 3 |

**Table A.1.** ML-Agents hyperparameters

**Listing 1.** NavGrid update pseudocode

```
def UpdateWeights(agent, action):
  c = agent.getClosestNode()
  c.setWeight(0)
  N = c.neighbors
  for n in N:
    dir = agent.getUnitVectorToNode(n)
    w = dot(action, dir)  # [-1, 1]
    n.addWeight(w)
    U = n.neighbors
    for u in U:
      if u in N:
        continue
      u.addWeight(w * discount)
```

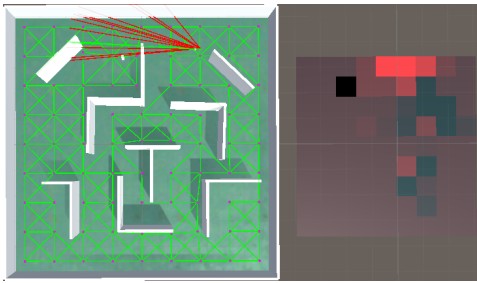

**Figure B.1.** An example `Navgrid` (left) and its weight visualized as a heatmap (right).

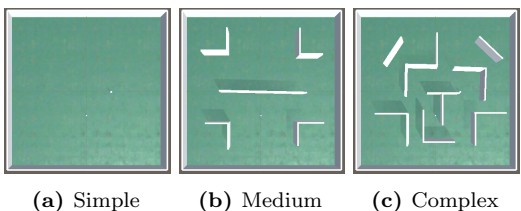

(a) Simple      (b) Medium      (c) Complex

**Figure B.2.** The audio environments of our experiment.

# B    Audio environments

In the experiment, we used three separate environments. They can be seen in Figure B.2. Surface properties are set as Steam Audio Geometry with the same absorption, scattering, and transmission values. Absorption affects how the low, medium and high frequencies are separately absorbed. Scattering affects the direction of reflection when sound is reflected from a surface. Transmission affects how much low, medium and high frequencies can pass through the surface without reflections.

