# OpenReview forum: "Towards concurrent real-time audio-aware agents with deep reinforcement learning"
_NLDL.org/2025/Conference — NLDL 2025 Oral_

### Official Review · Reviewer_PkUU · 2024-10-02

**Confidence:** 3

**Summary:**

This paper explores the use of deep reinforcement learning (DRL) to train audio-aware agents in video games, addressing the challenge of limited multi-listener support in game engines. The authors propose a distributed architecture where each client runs its own audio engine instance (Figure 1). They train and evaluate their approach using a hide-and-seek task, measuring performance in environments of varying complexity. Results suggest that their method enables the training of multiple concurrent audio-aware agents with good results, when compared to random baselines.

**Strengths:**

- Clear Presentation and Structure: The paper is well-organized and written in a clear style. The figures are informative and illustrate key findings. The background information, aka the problem at hand, is well explained, making the work accessible to readers with different backgrounds.

- The optimization task: The hide-and-seek approach is, as described in the manuscript, is clear and easy to understand. It follows popular ideas present in current self-supervised learning methods such as LLMs and vision models, where the model (agent) needs to predict an intentionally hidden property of the data (sound location).

- The evaluation benchmark: The authors study how the complexity of the audio environment (different levels of reflections and occlusions) affects the agent's ability to localize the sound source.

- Limitations: The authors honestly acknowledge the limitations of their work (e.g., reliance on external libraries, lack of faster-than-realtime audio) support.

Question: Could you elaborate on the rationale behind choosing the specific hide-and-seek scenario and the selected audio cues? I can envision many possibilities to increase the difficulty of the task, such as introducing a time constraint for the agent to find the sound and also making the target non-stationary. What are your insist about these?

**Weaknesses:**

- While the STFT-H agent appears to achieve slightly higher performance than the STFT-R agent (Figures 5, 6), there is a lack of discussion regarding possible reasons and implications of such results. Why does the STFT-H agent perform slightly better? Is it the simplicity bias from the STFT-R technique? Or is there something fundamentally beneficial in the Hanning window approach used to perform the STFT that collaborates with the results?

- The authors compare two variations of their proposed approaches with a RANDOM agent as a baseline. While the RANDOM agent serves as a reasonably good lower-bound performance, the comparisons lack an upper-bound performance which would demonstrate how well their approach compares with existing methods.

- Lack of Deep Reinforcement Learning Insights: Given the paper's title and focus on audio-aware agents, the analysis of the deep reinforcement learning is somewhat superficial. I missed a bit deeper exploration of the agents' learning dynamics, the impact of hyperparameter choices on the agent's behavior, or a justification for the PPO algorithm.

**Final Rebuttal Confidence:**

3

**Final Rebuttal Justification:**

Thank you for the response and additional clarification.

**Justification:**

The paper addresses a practical and relevant problem in game AI development and provides a functional solution. The experimental methodology is clear and well-defined, allowing for reproducibility. Overall, the paper offers valuable insights, positively contributing to the field.

---

> ### Author Rebuttal · Authors · 2024-10-25
>
> **We reacted to the weaknesses pointed out in the review in the following way:**
>
> - We added more description on the problem.
>
> - We added more description of our method.
>
> - We added more measurement results.
>
> It was pointed out that more insight into the machine learning could be given. We hope that the modifications made to the paper are sufficient to make the contribution of the paper clearer.
>
> **Our answers to the question presented by the reviewer:**
>
> The rationale behind choosing the specific hide-and-seek scenario was choosing a workload to the system that can be easily adjusted for different complexities. We fixed our wordings at the beginning of the paper as they may have been misleading: the focus of our contribution is not in the task itself (i.e, the navigation problem), but in the performance of the underlying system.

---

### Official Review · Reviewer_ZpAp · 2024-10-04

**Confidence:** 4

**Summary:**

This paper explores the application of reinforcement learning to train audio-aware agents that utilize both visual and auditory cues in real-time environments, particularly in a hide-and-seek scenario. The paper aims to overcome the limitations of game engines that typically support only a single audio listener by constructing a multi-listener based system. Through several experiments, the authors show the results on multi-listener performance and agent efficiency, while also measuring CPU and memory utilization to evaluate the feasibility of running multiple audio-aware agents simultaneously.

**Strengths:**

[Significance]
The authors construct a framework where multiple audio-aware AI agents can simultaneously process sound in a shared environment and ensure audio-agents can run in real-time. The authors also evaluate the performance of AI agents in tasks with various complexity from simple to complex. The results show that their AI agents outperform the random agents. The paper also compares the performance of AI agents trained by different STFT windowing functions which have some effects on the behaviors of the agents.

[Quality]
This paper provides the performance of the framework on the CPU and memory utilization, demonstrating its scalability. It can support up to 40 concurrent audio listeners with minimal performance degradation.

**Weaknesses:**

[Methods]
* The novelty of this paper is not enough. For example, compared to [1], both works present similar contributions in the domain of reinforcement learning with audio inputs. Both papers use similar techniques to integrate audio data into the reinforcement learning framework.

* It is unclear how the value loss is calculated, regarding whether the agent is rewarded for decreasing the length of the shortest path to the target.

* The procedure for updating the Navigrid is not described. For example, what is the formula for updating the weights in the Navigrid? What is the range for the weights of nodes in Navigrid? If a node propagates the increase in weights to its neighbors with a decay factor, why the weights around the agent are lower than the weights near the target in Figure B.1?

* In Appendix A (line 602), the paper mentions that "the target could be moving," and the GitHub README states, "Target-related scripts (Target of the audio agent, can move around randomly)." However, in line 254, it is mentioned that "the player is represented as a static object that does not move around in the environment." It is unclear how many targets are used in the environments and whether the targets move randomly during evaluation.

* The paper mentions the faster-than-realtime method in the methodology section, but it appears that it has not been implemented. It would be better to move it to the future work section.

[Experiments]
*  The paper does not have comparisons with other established baselines in the field of reinforcement learning with audio inputs.

* The paper does not include any ablation studies that examine the performance of the agent when trained solely on positional data and ray tracking, without incorporating audio data. As a result, it remains unclear whether adding audio information is necessary for this task.

* The README file on GitHub does not provide sufficient detail regarding the setup and execution of the code, which impedes the ability of other researchers to reproduce the experimental results reported in the paper.

Questions for authors:
1. The approach involves reinforcement learning training, which typically needs GPU resources. What are the specifications for GPU? If GPU resources are required, would training multiple agents simultaneously cost a lot of computation resources? Could you provide details on how resource usage is managed during training?
2. In section 3.4, it mentions head related transfer function (HRTF), which models the interaural level differences (ILD) and the interaural time difference (ITD). Is this function also used in the framework?
3. In Section 3.5, does this paper employ the faster-than-realtime technique as described in [1]?
4. Section 4.1 mentions that each Unity instance is running the exact same scene with Steam Audio and one audio agent with a CNN-based audio model. Do multiple agents train by the shared training data collected by all agents, or does each agent only trained by its own training data?
5. Can the agent successfully locate the target if it is trained solely using audio data?
6. Figure 5 shows the training curves obtained with 10 concurrent Unity instances. Do the results in Figure 6 use the same agents that were trained in Figure 5?
7. In Section A.2, do the random agents also use the Navigrid? If they do not use the Navigrid, would it be unfair to compare their performance with that of the trained agents?

[1] Hegde, A. Kanervisto, and A. Petrenko. “Agents that listen: High-throughput reinforcement learning with multiple sensory systems”. In: 2021 IEEE Conference on Games (CoG). IEEE, Aug. 2021. doi: 10.1109/cog52621. 2021.9619096.

**Final Rebuttal Confidence:**

3

**Final Rebuttal Justification:**

The authors have addressed the main concerns, particularly the motivation of this paper. The primary focus of this paper is on the parallelization of the multi-listener system, rather than advancing methodologies for audio-aware agents. The revision provides several new experiments with detailed explanations. Figure 6 also demonstrates the benefits of running multiple Unity instances concurrently. In conclusion, although the novelty of this paper is limited, the authors have effectively addressed most of my concerns, and I am inclined to accept this paper.

**Justification:**

In summary, this paper lacks novelty in the methodology of reinforcement learning related to audio. In addition, the paper falls short of providing detailed methods, affecting the overall clarity and reproducibility of its contributions. Please refer Weaknesses for details.

---

> ### Author Rebuttal · Authors · 2024-10-25
>
> **We reacted to the weaknesses pointed out in the review in the following way:**
>
> 1) Overall, we clarified the paper contribution by modifying the wordings in the paper abstract and introduction that may have been misleading. As the focus and novelty of the paper is not in the use of audio-aware agents as such, but in the parallelization of the multi-listener system and the related learning performance of individual agents, we also made the following modifications:
>
>     - We added more measurements and explanation to better describe the behavior.
>
>     - We clarified the problem description to explain how the multi-listener and parallelization parts are linked.
>
>     - We added more explanation to the results section to make the thematic flow of the paper better.
>
> 2) We clarified the description, and the measurement results related to the reward signal.
>
> 3) Our paper was unclear on the Navgrid concept and the related weigths. Specifically, the weights are lower next to the agent, because the weight is set to zero for the current location (i.e., we can be confident that the target is not here). To make the Navgrid concept and the related weights clearer, we added more description to the appendix.
>
> 4) We clarified the issue on static versus moving targets.
>
> 5) Previously the paper gave the impression that we had implemented faster-than-realtime operation, which we have not done. Our intention was to point out that while faster-than-realtime operation is often used to speed-up the training of DRL-agents, using audio as an observation makes faster-than-realtime more difficult to implement. We clarified the related parts of our paper.
>
> 6) We added more views into the related literature.
>
> 7) The review points out that the paper does not include any ablation studies that examine the performance of the agent when trained solely on positional data and ray tracking, without incorporating audio data.
>
>      -   We see that this relates to the overall contribution of the paper. Our intention was not to study the navigation as such but the multi-listener performance and the related runtime performance. We made several modifications to the paper to clarify this (see item 1). In order to more clearly focus on the audio observations in our experiment, we removed the positional data and ray tracing from the agent inputs.
>
> 8) We improved the README description in the related Github repository.
>
> **Our answers to the questions:**
>
> 1) Our previous measurements were using only CPU. We added results on GPU versus CPU performance and the related specifications. In general, using GPU yields faster SGD times, but otherwise the good operation point (especially considering the number of environments used) is roughly the same for both (using GPU versus CPU only).
>
> 2) The usage of head related transfer function (HRTF) was unclear. And yes, it is used. HRTF is enabled with Steam Audio using the default parameters.
>
> 3) Our experiment does not include faster-than-realtime operation (see our reaction to weakness 5). We did not implement such operation, but we identify it is an important feature for training and evaluating audio-aware agents.
>
> 4) It was unclear if multiple agents train by the shared training data collected by all agents, or does each agent only get trained by its own training data.
>
>    - Data from all agents is gathered into a shared buffer, which is then used to optimize the model, e.g., compute new weights with the SGD following PPO.
>
> 5) In order to clarify the contribution, we modified the experiment to include only audio agents.
>
> 6) The results in Figures 7, 8, and 9 (there was only Figure 6 in our original versio) use the same agents that were trained in Figure 5. Figure 5 is now also averaged over 3 training seeds instead of 1.
>
> 7) It was unclear, do the random agents also use the Navigrid.
>
>     - Description of the random agents was clarified. Yes, they do use an approach similar to the Navgrid and yes it would be unfair otherwise.

---

### Official Review · Reviewer_6bjW · 2024-10-09
**Paper showing why concurrent audio aware agents would be useful in a game engine setup and how DRL is useful in building such agents.**

**Confidence:** 4

**Summary:**

The paper focuses on enabling deep reinforcement learning agents (PPO Based) to use audio cues for navigation. The authors demonstrate a method for incorporating multiple concurrent audio-aware agents in a game engine (Unity) by using a multi-client approach, overcoming the common limitation of supporting only one audio listener per game instance.

The paper provides good motivation for the problem and shows details on the settings considered (along with images of the environment settings) and explanations on NavGrid.

**Strengths:**

1) Clear motivation - on why concurrent audio-aware agents would be useful in game engine setup like Unity
2) Combining RL with Audio Cues - Using Audio specific sensors along with other state representations is a novel contritbution - making it a strength of the work
3) The authors use different environment setups (across varying difficulty levels) to show how their approach out-perfoms a random agent
4) The work has a practical application in game engines and the authors also discuss about the limitations and some potential future works which is good to see.

**Weaknesses:**

1) Lack of gathering results over multiple seed runs - RL typically atleast 3 seed runs are considered necessary. (even though we see a convergence behavior in training pretty fast)
2) Have the authors tested the SPL score without the audio aware agents and other than random? its a bit unclear how those would perform
3) Plots could be made more clearer - eg Fig 3 by showing units of throughput (or defining it before) and Fig 6 - things like how many agents was used in that experiment are necessary (bit unclear)
4) Maybe more explanation of why SPL is the right evaluation metric in this case would be good to add as well. Seems like the paper over relies on this single metric? Some other analysis on agent behaviors would be interesting.
5) Would this behavior also extend to complex scenarios where the player is not stationary and more dynamic ( as in most realistic scenarios?) Has any experiment been done in this direction?

**Justification:**

Confident that this type of work puts in a different perspective for the use of audio cues as an input modality for RL agents - is especially useful in game engine setups and the authors show how concurrent audio aware agents could be designed.

---

> ### Author Rebuttal · Authors · 2024-10-25
>
> We reacted to the review in the following way:
>
> 1. We included results from 3 seeded runs into the paper.
>
> 2. In order to clarify the contribution, we modified the experiment to include only audio agents (that have no other sensors than the audio sensor). The motivation of using random agents was to give the readers some comparison, not to set bounds (etc.). We tried to clarify this.
>
> 3. We added better labeling to the plots and clarified the textual explanation.
>
> 4. We clarified the SPL explanation, but included also results with other metrics.
>
> 5. We designed the experiment code to support dynamically moving targets, but we did not include measurements with moving targets in this paper. This is left as a future work. Based on preliminary measurements, a dynamic target might even help the agent if it moves out of a location that is otherwise difficult to locate. However, more measurements would be required to make more extensive conclusions. An even more interesting scenario would be to have a target that delibirately attempts to hide from the agent.

---

### Official Review · Reviewer_ft3k · 2024-10-09

**Confidence:** 2

**Summary:**

The paper describes the design of audio-based deep rl agents, in a multi-listener system.

The paper tackles a navigation problem: in a game engine, an agent tries to find the source of a noise. This is formalized as an RL problem, where the observations are visuals and sounds, the actions are simple navigation capabilities, and the reward is the opposite of the target distance. The authors propose to enable a multi-listener support, which allow them to parrallelize the learning procedure.

The performance of the audio server and the agents is then evaluated in the Unity engine.

**Strengths:**

The approach of the paper is reasonable: it proposes a sound solution to the multi-listener problem, and uses an simple but effective approach to train  the DRL agents.

**Weaknesses:**

My main concern with the paper is that the contributions are not very clear. In particular, the paper introduces two contributions, the multi-listener / parallelization part, and the DRL for navugation part, but is not clear how they are linked. I think what could improve this is the following/

First, it would be interesting to see the impact of the number of listeners on the evloution of the reward. I assume the most listener there are, the faster the reward increases ? This could be dicussed or shown as an experiment.

Second, and more generally, there could be a more explicit discussion on what is the link between these two contributions. Is the goal that more listeners will improve sample efficiency? IS the goal being able to develop multi-agent algorithms?

**Justification:**

Overall, I find the paper to be of good quality.

I think it lacks a bit of clarity, not in the techniques presentation, but on the problems it try to solve.

---

> ### Author Rebuttal · Authors · 2024-10-25
>
> We reacted to the review in the following way:
>
> * We clarified the paper contribution by modifying the wordings in the paper abstract and introduction that may have been misleading.
>
> * To address the pointed weaknesses of the linkage between the multi-listener and training parts we made the following modifications:
>
>    * We added more measurements to show how our method performs.
>
>    * We clarified the problem description to explain how the multi-listener and training parts are linked.
>
>    * We added more explanation to the results section to make the flow of the paper better.

---

### Meta-Review · Area_Chair_bUUd · 2024-11-01

**Recommendation:** Accept (Oral)
**Confidence:** 4

**Metareview:**

The paper addresses the (sometimes) overlooked area of audio based RL agents. The authors propose a solution where they focus on the parallelization of multi-listener systems, with a complete and well-motivated set of experiments. The work is both relevant and timely and can generate interesting discussions within the community.

All the reviewers agree on the value of the contribution. The authors have also carefully considered their suggestions and improved the paper accordingly.

**Suggested Changes To The Recommendation:**

2: I'm certain of the recommendation.  It should not be changed

---

### Decision · Program_Chairs · 2024-11-06

**Decision:**

Accept (Oral)

**Comment:**

We recommend an oral and a poster presentation given the AC and reviewers recommendations.